# Sandwich Hybridization Assay for In Situ Real-Time Cyanobacterial Detection and Monitoring: A Review

**DOI:** 10.3390/bios12080640

**Published:** 2022-08-14

**Authors:** Ping Gong, Anna K. Antrim, Sarah R. Bickman, Emily G. Cooley, Seung Ho Chung

**Affiliations:** 1Environmental Laboratory, U.S. Army Engineer Research and Development Center, Vicksburg, MS 39180, USA; 2Oak Ridge Institute for Science and Education, Oak Ridge, TN 37830, USA; 3LightDeck Diagnostics, Inc., Boulder, CO 80303, USA; 4Bennett Aerospace, Inc., Raleigh, NC 27603, USA

**Keywords:** cyanobacteria, harmful algal bloom (HAB), sandwich hybridization assay (SHA), nucleic acids, amplification-free, real-time in-situ monitoring, water quality, public health

## Abstract

As cyanobacterial harmful algal bloom (cHAB) events increase in scale, severity, frequency, and duration around the world, rapid and accurate monitoring and characterization tools have become critically essential for regulatory and management decision-making. The composition of cHAB-forming cyanobacteria community can change significantly over time and space and be altered by sample preservation and transportation, making in situ monitoring necessary to obtain real-time and localized information. Sandwich hybridization assay (SHA) utilizes capture oligonucleotide probes for sensitive detection of target-specific nucleic acid sequences. As an amplification-free molecular biology technology, SHA can be adapted for *in-situ*, real-time or near real-time detection and qualitatively or semi-quantitatively monitoring of cHAB-forming cyanobacteria, owing to its characteristics such as being rapid, portable, inexpensive, and amenable to automation, high sensitivity, specificity and robustness, and multiplexing (i.e., detecting multiple targets simultaneously). Despite its successful application in the monitoring of marine and freshwater phytoplankton, there is still room for improvement. The ability to identify a cHAB community rapidly would decrease delays in cyanotoxin analyses, reduce costs, and increase sample throughput, allowing for timely actions to improve environmental and human health and the understanding of short- and long-term bloom dynamics. Real-time detection and quantitation of HAB-forming cyanobacteria is essential for improving environmental and public health and reducing associated costs. We review and propose to apply SHA for in situ cHABs monitoring.

## 1. Introduction

Harmful algal blooms (HABs) are characterized by increased phytoplankton biomass, declined dissolved oxygen, and sometimes by the production of cyanotoxins [1]. In freshwater, HABs tend to be dominated by cyanobacteria. Cyanobacterial harmful algal blooms (cHABs) are becoming more common around the world due to excess nutrient loads, eutrophication, and the changing climate [2]. cHABs, particularly *Microcystis*-dominated blooms, have been observed in every continent except Antarctica and have significant economic consequences. For example, 20 years ago, cyanobacterial blooms caused annual economic losses of up to $82 million on public health, fishery, and tourism in the U.S. alone [3]. cHABs are formed by a large number of genera, including but not limited to *Microcystis*, *Anabaena/Dolichospermum*, *Aphanizomenon*, *Cylindrospermopsis*, *Planktothrix, Lyngbya/Microseira,* and *Phormidium/Microcoleus*.

It has been extensively reported that cHAB-forming cyanobacteria can produce hundreds of metabolites known as cyanotoxins that are harmful to exposed aquatic and terrestrial animals, including humans [1,4]. For instance, *Microcystis* spp., *Anabaena/Dolichospermum* spp., and *Planktothrix* spp. are able to synthesize the peptide hepatotoxins, microcystins [5]. *Cylindrospermopsis* spp., *Aphanizomenon* spp., *Anabaena* spp., and *Lyngbya wollei* can produce hepatotoxic cylindrospermopsins [6]. Neurotoxic saxitoxins can be produced by *Anabaena/Dolichospermum* spp., *Aphanizomenon* spp., *Cylindrospermopsis raciborskii*, *Planktothrix* spp., and *Lyngbya wollei* [6,7]. Anatoxins, another class of neurotoxins, are produced by *Aphanizomenon* spp., *Anabaena/Dolichospermum* spp., and *Phormidium* spp. [8]. Notably, some of these genera, such as *Anabaena/Dolichospermum* spp., have the potential to produce all common classes of cyanotoxins. Although cyanotoxins-coding genes are often associated with HABs events, the detection of such genes does not explicitly imply the presence of cyanotoxins in the bloom [9].

A wide variety of approaches and technologies have been developed for the detection and monitoring of HABs-forming cyanobacteria, ranging from microscopic enumeration, analysis of Chlorophyll a, ATP and phycocyanin, quantitative polymerase chain reaction (qPCR), next-generation sequencing (NGS), enzyme-linked immunosorbent assays (ELISA), and high-pressure liquid chromatography (HPLC), to hyperspectral imaging, remote sensing, automated cell imaging systems, and machine learning [10]. When HABs occur, there are many questions, including whether a HAB event has occurred, what cyanobacteria are present and caused the event, whether and what cyanotoxins are released, and how dynamic the cyanobacteria and cyanotoxins exist. Many tools are necessary to answer these questions, and there is currently a lack of portable and field-deployable tools for in situ real-time identification and monitoring of cyanobacteria in the waterbody of concern. In situ, real-time monitoring is preferred, since there is a loss of sample representation during sample collection, transportation, and storage [11]; and cyanobacterial community composition can change spatially and temporally. Therefore, there is a need for rapid, real-time, and reliable tests that can provide local or in situ answers for cyanobacterial monitoring.

Here we introduce sandwich hybridization assay (SHA) as a cost-effective and accurate real-time, field-portable tool to identify and quantify cHAB-forming cyanobacteria. We begin with a brief review of SHA development history, summarize and compare different variations of SHA, then discuss its current applications to cHAB monitoring with an emphasis on pros and cons, and finally conclude with remarks on future perspectives.

## 2. Sandwich Hybridization Assay (SHA)

### 2.1. What Is a SHA?

Hybridization is a process of two complementary single-stranded DNA or RNA molecules forming a single double-stranded molecule through base pairing. Sandwich hybridization assay (SHA) is a molecular technique based on successive hybridization of two oligonucleotide probes: a capture probe used to immobilize the target DNA or RNA on a solid support and a signal probe labeled with a detectable marker to quantify the target copy number (see Figure 1) [12]. A segment of the target molecule is “sandwiched” between the two probes in this amplification-free, direct capture method [13]. In general, the capture probe is designed with a high specificity to the target cyanobacteria, whereas the signal probe hybridizes to a conserved sequence found in target gene and is often labeled with either a fluorophore or a digoxygenin (DIG) to obtain a measurable signal. Considerations in capture probe design include but are not limited to the following: specificity to target organisms, proximity to signal probe (within 100–250 bp), <70% similarity with signal probe, GC content (i.e., percentage of guanine and cytosine in a DNA or RNA molecule) between 40% and 60%, melting temperature between 69 °C and 74 °C, secondary structure stability less than 34 °C, and homodimer stability less than 17 °C [14]. When a signal probe is labeled with a fluorophore (e.g., Cy5), a fluorescence reader is used to read out the target molecules trapped by the capture probe [15,16]. For DIG-labeled signal probes, anti-DIG antibodies in conjugation with horseradish peroxide (HRP) or alkaline phosphatase (AP) are added along with such a substrate as 3,3′,5,5′-tetra-methylbenzidineto (TMB) or 2′-(2-benzothiazolyl)-6′-hydroxybenzothiazole phosphate (BBTP) to produce a colorimetric or fluorescent readout [12,14,17].

### 2.2. SHA Development and Application

The SHA was initially developed in the late 1970s for mapping transcripts to the genome of adenovirus type 2 [18]. It was nearly twenty years later when the first U.S. patent for this technology was awarded in 1995 to Jennifer K. Ishii and Soumitra Ghosh, both with Siska Diagnostics Inc., La Jolla, CA. The two inventors developed a two-step sandwich hybridization technique for detection of as little as 10^−17^ moles of nucleic acid molecules in solution, without requiring the use of radioactive compounds: the target nucleic acids are captured by hybridization with oligonucleotides covalently attached to a polystyrene solid support to form complexes that are then hybridized to detection oligonucleotides [19]. Three years later, Tyagi et al. [20] were granted another U.S. patent for multiple SHA background “noise” reduction measures, such as the use of separate capture and signal/reporter probes, separation from immobilized capture probes by cleavage and isolation, use of RNA binary probes and an RNA-directed RNA ligase, and amplification by an RNA directed RNA polymerase.

Table 1 summarizes some representative applications and methodological alterations and improvements as reported in the literature. The SHA has been applied to the detection of target RNA or DNA molecules in a wide variety of prokaryotic and eukaryotic organisms. Prokaryotes include but are not limited to adenovirus [18], human cytomegalovirus [21], *Escherichia coli* [17], lactic acid bacteria [22], and pathogenic bacteria (e.g., *Salmonella* spp. [23], *Legionella* spp. [24], and *Bacillus* anthracis [25]). Eukaryotes include yeast [12], marine invertebrate larvae [14], chamois [26], and particularly microscopic, single-celled, and HAB-causing non-cyanobacterial phytoplankton, such as marine diatom *Pseudo-nitzschia* spp. [27,28,29,30,31], marine eukaryotic alga *Heterosigma akashiwo* [32,33], and marine dinoflagellates, such as *Karenia brevis* [34], *Alexandrium catenella* [30,35], *Cochlodinium polykrikoides* [36], and *Fibrocapsa japonica* [32]. The following are a few examples of SHA applications. First, the SHA technique was used in detecting the 23S rRNA gene in *Salmonella* spp. by combining the use of microtiter plates and a visual colorimetric reaction [23]. Since *Salmonella* is a widespread pathogen for both humans and animals, rapid detection can be critical for diagnosis and treatment of food and water-borne outbreaks. This microtiter plate SHA for *Salmonella* was found to be rapid (completion within 3 h), sensitive, and specific (detecting pure cultured *Salmonella* and non-*Salmonella* bacteria strains with a zero-error rate, as well as 98.2% of positive natural samples and 99.5% of negative natural samples), with a detection limit of 1.8 × 10^5^ colony forming units (cfu)/mL [23]. Second, Leskelä et al. [24] applied SHA to the detection of *Legionella* spp. and achieved a detection limit of 2 × 10^−17^ mol of target molecules, corresponding to 1.2 × 10^7^ molecules of 16S rRNA or approximately 1800 *Legionella* cells, by using a 3′-end biotin-labelled capture probe and a 3′-end DIG tailed detection probe. Thirdly, SHA was used to detect sequences as short as 22-nucleotide DNA fragments [15] and microRNAs (miRNAs) [16] and to investigate ancient DNA in the fields of evolutionary biology and molecular ecology, as well as altered DNA in food fraud detection and forensics using nucleic probe labeled with rhodamine 6G that enables the Surface Enhanced Resonance Raman Scattering (SERRS) technology for specific DNA detection [26].

The SHA protocol has been modified to meet specific requirements for a wide range of application. For instance, the solid support has evolved from membrane materials in early times [37] to nylon beads [27,28], magnetic beads [12,17,21,26], polystyrene prongs [14,19], microarray glass slide [16], and microtiter plates [21,25]. Due to their physical porous structure, membranes give high background and steric constraints, causing difficulties in quantitative analysis [21,37]. Beads possess a large surface area with a well-defined capacity and rapid binding kinetics, leading to a higher fixation capacity and a higher hybridization yield, whereas microtiter plates and microarrays are better adapted for the simultaneous handling of a large number of samples [16,21,25]. In order to immobilize a capture probe to a solid support, the support surface may be coated with streptavidin to which the biotinylated capture probe is bound [12,14,17,26,27,28]. Alternatively, the capture probe can covalently bind to aminated solid surface via a carbodiimide crosslinker [16,21,25]. The use of unlabeled “helper” probes (between capture and signal probes) increased hybridization efficiency by 15- to 40-fold [17,38]. Instead of Cy5-, Cy3-, or DIG-labelling at one end, double DIG-labeling at both the 5′- and 3′-ends of detection probes significantly enhanced signal intensity [14].

The synthetic DNA mimic peptide nucleic acid (PNA) have been used to replace DNA probes [15,25] because of its superior hybridization characteristics and improved biochemical properties, including resistance to enzymatic degradation, increased sequence specificity to complementary DNA, and higher stability when bound with complementary DNA [25,39]. Although the reported limit of detection (LOD) for miRNA using PNA probes was 10 nM corresponding to 2 × 10^11^ target molecules in a 30 μL sample vial (at least 200-fold higher than using DNA probes, see Table 1 for more), million-fold increases in target concentration can be realized to achieve the theoretically ideal LOD for low abundance miRNA targets on the order of 10^4^ targets, 7 orders of magnitude lower than the reported LOD [15]. For non-miRNA targets, a cyclopentane-modified capture PNA probe (PNAα) in combination with a biotin-labeled signal PNA probe (PNAβ), a commercially available polymer of Horse Radish Peroxidase-avidin (poly-HRP-avidin) and tetramethylbenzidine (TMB), can detect 10 zmol (1 zeptomole = 10^−21^ mole or 10^−6^ femtomole (fmole)) of target DNA [25], which is 3000-fold lower than the 0.03 fmole of LOD [16,21], the lowest that we are aware of for DNA probes.

From the instrumentation perspective, a regular SHA would not require anything but visual inspection if a qualitative endpoint of color change is measured or would simply use a plate reader that measures quantitatively either fluorescence (e.g., at excitation wavelength 430 nm and emission wavelength 560 nm) [12,17] or absorbance (e.g., optical density at 450 nm) [14,21,25]. Other sophisticated equipment, such as liquid scintillation counter and SERRS, have been employed when the detection probe is labeled with [^35^S]-ddATP [21] and rhodamine 6G dye [26], respectively. A capillary electrophoresis running nonionic surfactant micelle-containing buffers was used to separate the sandwich complex (i.e., a target DNA sandwich-hybridized with a γ-substituted PNA amphiphile (γPNAA) probe and a DNA probe) from unbound γPNAA probes, DNA probes and target DNAs via a mobility shift assay [15]. A microarray scanner is needed when capture probes are printed on a glass slide [16].

Typical SHA protocols can be completed within 3–4 h (see Table 1) and often use saline-sodium citrate (SSC, e.g., 1× SSC made of 0.15 M NaCl and 0.015 M sodium citrate) as the main component of hybridization and washing buffers. Compared with SSC/NaCl-based buffers, GuSCN (guanidine thiocyanate) is another commonly used base reagent in hybridization buffers [14,27,28] because it is effective at disrupting cells, inactivates nucleases, and permits direct, specific hybridization at much lower temperatures [40,41]. Other widely used ingredients in these buffers include Tween 20, polyvinyl pyrrolidone (PVP), ethylenediaminetetraacetic acid (EDTA), sodium dodecyl sulfate (SDS), Tris, formamide, maleic acid, and dextran sulfate (see Table 1).

### 2.3. SHA Application to Cyanobacterial Detection and Monitoring

Although SHA has been around for nearly five decades, its application to cHABs detection and monitoring, to the best of our knowledge, began only two decades ago, which was likely driven by the rising demand for molecular technologies enabling in situ identifying, sensing, and monitoring HAB-causing cyanobacteria [42]. Our literature survey identified seven peer-reviewed papers and one thesis publication (see Table 2 for summary). Matsunaga et al. [43,44] designed the first set of capture probes targeting the genus specific region of the 16S rRNA sequences from five cyanobacterial genera (*Anabaena*, *Microcystis*, *Nostoc*, *Oscillatoria*, and *Synechococcus*). These probes were immobilized on bacterial magnetic particles (BMPs) isolated from the magnetic bacterium *Magnetospirillum magneticum* AMB-1 via streptavidin-biotin conjugation. A DIG-labeled cyanobacterial universal probe CYA781R was used as the signal probe [45]. An anti-DIG-AP antibody was used for signal detection after addition of the CDP-Star™ Substrate with Emerald-II™ Enhancer or the AttoPhos^®^ AP substrate. Results demonstrated high discriminatory power of the genus-specific capture probes, which produced significantly higher fluorescence when hybridized to the 16S rRNA amplicons from the strains belonging to their respective target genus [43]. The authors further automated the entire hybridization and detection process using a magnetic separation robot and transformed the assay into a 96-microwell format to increase the throughput [44].

Castiglioni et al. [46] developed a microarray spotted with universal oligo probes (i.e., 5′ NH_2_-modified “zip code” oligonucleotides carrying a poly (dA)_10_ tail at their 5′ ends covalently immobilized on a CodeLink slide) to profile the abundance and diversity of 19 cyanobacterial groups identified by phylogenetic analysis of 338 sequences of cyanobacterial 16S rRNA genes. Prior to array hybridization, a 30-cycle sandwich hybridization (called ligation detection reaction, LDR) was performed in a thermal cycler with an LDR mixture made of a discriminating probe labeled with Cy3 dye at the 5′ end, a common probe phosphorylated at the 5′ end and carrying a czip code (i.e., oligos complementary to the “zip code” probe) at the 3′ end, a DNA ligase, and a purified PCR product of cyanobacterial 16S rRNA. Each pair of discriminating probe and common probe (excluding the czip code) was designed specifically to target one of the 19 cyanobacterial groups. Array hybridization was carried out in a dark chamber at 65 °C for 1 h. After post-hybridization washing and drying, the Cy3 green fluorescence intensity was acquired for array spots using a laser scanner with settings of λ_ex_ = 543 nm and λ_em_ = 570 nm. This SHA-based microarray approach was validated by testing 95 known 16S rRNA amplicons of single strain (24 from axenic strains, 27 from isolated strains, and 44 from cloned fragments recovered from lake samples), unbalanced mixtures of different known 16S rRNA amplicons, and an unknown environmental sample, all of which demonstrated a high discriminative power and sensitivity (LOD = 1 fmole).

The above-mentioned three early studies only evaluated PCR products of 16S rRNA in pure cultured or environmental cyanobacteria. Later studies directly analyzed non-amplified nucleic acids extracted from pure cyanobacterial cultures or environmental samples. Zhu et al. [47,48] designed two pairs of *Microcystis*-specific capture and signal probes, one targeting the PC-IGS (phycocyanin intergenic spacer) region [49] and the other targeting the *mcyJ* gene. These probes not only qualitatively discriminated *Microcystis* from other cyanobacterial genera (*Anabaena*, *Aphanizomen*, and *Planktothrix* (*Oscillatoria*)) but also quantitatively detected *Microcystis* populations at environmentally relevant densities as low as 100 cells/mL. These SHA results were validated by microscopic enumeration technique [47,48]. Following the method of Goffredi et al. [14], Dearth and coworkers [50,51] also designed a *Microcystis*-specific capture probe (MIC593) and adopted the bacterial universal probe EUB338 [52] as the signal probe, both targeting the 16S rRNA gene. They modified the Goffredi method [14] by replacing the biotin-coated polystyrene prongs with a streptavidin-coated microwell plate, and the streptavidin-biotinylated capture probe with biotinylated capture probe. The modified SHA had a LOD of 1.5 × 10^4^ cells/250 μL homogenate and a linear range between 7.75 × 10^4^ and 1.30 × 10^6^ cells/250 μL homogenate, corresponding to the cell density of a moderate *Microcystis* bloom (1.00 × 10^5^ cells/mL). Using the modified method, the authors investigated the influence of environmental factors (light intensity and temperature) on *Microcystis* populations.

Another major development in this field is the integration of SHA to an automated sampler such as Environmental Sample Processor (ESP). Motivated by the so-called “ecogenomic sensors” notion of using an ordered array of different probes to detect a variety of organisms in a single sample, a group of researchers in the Monterey Bay Aquarium Research Institute (MBARI) developed a novel SHA-based probe array as one of the analytical modules in the ESP [53]. A series of publications by this group documented the conceptualization, development, field demonstration, deployment, and refinement of ESP to meet the growing needs of in situ, real-time HABs investigation [29,30,31,41,42,54]. One of these publications [41] reported the use of target-specific SHA probes for successful detection of 16S rRNA indicative of marine cyanobacteria (*Synechococcus*) and other phylogenetically distinct clades of marine bacterioplankton in a 96-well plate format as well as low-density ESP arrays printed on a membrane support. Samples subjected to investigation included target and non-target products derived from in vitro transcription of 16S rRNA genes as well as extracted RNA from collected natural seawater. Reported detection limits were between 0.10–1.98 and 4.43–12.54 fmole/mL homogenate for the 96-well plate and array SHA, respectively. Furthermore, this study demonstrated that marine bacterioplankton (including cyanobacteria) can be assessed remotely, in situ, using SHA probe arrays integrated in the ESP.

More recently, Microbia Environment Inc. developed and patented CARLA (Cellular Activity RNA-based eLisA), a commercial biosensor technology based on the SHA with a detection probe coupled with an enzymatic activity that induces a colorimetric signal proportional to the quantity of sampled rRNA (https://www.microbia-environnement.com/en/technology/) (accessed on 20 July 2022). Species-specific detection and quantification of target cyanobacteria, such as *Microcystis*, *Planktothrix*, and the ADA clade (*Anabaena*/*Dolichospermum*/*Aphanizomenon*), can be accomplished in less than 3 h after RNA extraction from water samples.

### 2.4. Advantages of SHA

When used as a molecular tool for routine detection and monitoring of HAB-causing cyanobacteria, SHA has many advantages over conventional techniques such as light microscopy and quantitative polymerase chain reaction (qPCR). Traditionally, samples were collected and transported to a laboratory where microscopy was performed to identify and enumerate cyanobacterial communities. The performer requires special training and hands-on experiences in morphological observation and taxonomic identification or classification. This process is time-consuming (taking days or longer) and labor-intensive, and it may introduce personal bias. Even though such automated and field deployable cell imaging systems as Imaging Flow CytoBot (IFCB, https://mclanelabs.com/imaging-flowcytobot/) (accessed on 20 July 2022) and FlowCAM (https://www.fluidimaging.com/) (accessed on 20 July 2022) can quickly, accurately, and reliably identify and quantify cyanobacteria, they are currently cost-prohibitive, limiting their wider application for in situ detection and monitoring of cHAB species. In contrast, SHA is convenient to perform and appears to be rapid (a few hours), accurate (no or low cross-reaction with non-target species), repeatable, reliable, and highly amenable to automation (using robotic workstation) and multiplexing (in 96-well or array format) [41,44,46,50,51].

Although qPCR is more sensitive with a much lower detection limit (a few copies of target molecules) [55] and a much broader linear dynamic range (>5 orders of magnitude) [56], SHA does not require amplification and sophisticated equipment, and its readout may be visualized by color change or quantified using smaller, more portable, and much less expensive instrumentation, making it more cost-effective and field-deployable to track target species in near real-time and in situ [29,30,31,41,54]. Although new omics technologies (i.e., metagenomics, transcriptomics, proteomics, and metabolomics) are fast-growing and powerful tools with great potential applications for cyanobacterial community diversity and dynamics studies [57], such applications are currently limited to laboratory benchtop research [42]. In a comparative study for quantifying laboratory cultures of the ichthyotoxic raphidophyte *Heterosigma akashiwo*, Doll et al. [33]. observed a high degree of correlation between qPCR and SHA responses. With an LOD at 1 fmole of target molecules [46] or 100 cells/mL [47,48], SHA possesses a sensitivity required for detecting typical cyanobacterial populations observed in the early phase of blooms.

Other benefits of using SHA over other molecular methods include relatively low per sample cost, processing time, and capital investment on instrumentation. When integrated into an ESP, the cost of SHA is estimated to be between $7 and $10 per sample [53]. The hands-on time spent performing SHA is estimated to be 15–20 min and approximately 30–40 samples can be analyzed in an eight-hour day, when using an automated processor.

### 2.5. Technical Limitations of SHA

SHA is often considered a semi-quantitative assay [53] with a narrow linear range of two orders of magnitude, e.g., 5 × 10^2^ to 2.5 × 10^4^ cells/mL [47] or 1–100 pM of target nucleic acids [16]. Although the capture probes are capable of distinguishing between target sequences with as little a difference as a single base pair [15], such discriminatory power is dependent on the availability of sequence information on taxa of interest. This is mostly a concern in developing capture probes for a genus or a phylogenetic clade, which often target such ribosomal subunits as 16S, 18S, 23S, and 28S rRNAs. These rRNAs are highly conserved such that capture probes targeting these sequences risk cross-species hybridization and cannot separate closely related species of interest. A solution to this issue is to select unique or divergent genes in the target organism (e.g., *mcyJ* gene from *Microcystis* strains [47,48]). Moreover, SHA measures the copy numbers of target genes (e.g., 16S rRNA), which cannot be directly converted to cell density of target cyanobacterial strains without constructing a standard curve between cell counts and signal intensity or gene copy number. This is due to the fact that the expression level of assayed gene (copy number per gene) is affected by the cellular physiological status [33,51]. SHA results can be adjusted accordingly, when it is known how many copies of 16S rRNA are present in a cyanobacterial cell (e.g., *Microcystis aeruginosa* NIES-843 [58]).

Similar to other amplification- or hybridization-based techniques (e.g., regular PCR, qPCR, reverse transcription PCR, southern blotting, and northern blotting), cell lysis is an important preparatory step in SHA to release target intracellular molecules. The sensitivity and quantification accuracy of SHA are affected by the quality and yield of extracted nucleic acids. For cyanobacteria, some species are easier to lyse than others. For instance, *Lyngbya* spp. are typically difficult to lyse due to thicker cell walls and protective sheaths [59]. Many physical, chemical, and enzymatic methods have been employed to lyse cyanobacterial cells, however, there is no one method capable of disrupting all cyanobacterial species in a satisfactory fashion [60]. For instance, it was reported that xanthogenate, a polysaccharide solubilizing compound, was able to lyse a wide variety of cyanobacterial genera but with varying RNA yields [60]. Furthermore, xanthogenate and other common chemicals used for lysis may interfere with SHA reagents, leading to inaccurate results [61]. Although chemical lysis could cause interference with the SHA chemistry, repeated freeze-thaw cycles or mechanical bead beating in combination with a lysozyme can lead to more efficient RNA extraction [50,51,59,62].

## 3. Future Perspectives

### 3.1. Does SHA Have an Established Niche in Field Work for cHABs?

As a fast, sensitive, and probe sequence-specific molecular technique, there exist at least three different scenarios where SHA can play a unique role for in situ and near real-time detection of HAB-associated cyanobacteria in field settings. First, these assays are useful in field surveys where discrete and spotty samples are collected and near real-time answers are expected for the presence and abundance of specific cyanobacterial species or genus suspects. Second, they can be used for routine monitoring of specific field sites where discrete samples are collected and near real-time detection is required for a list of specific cyanobacterial species or genera. Third, they provide key information when continuous data are collected for monitoring for a specific array of HABs species (including cyanobacteria). For the first two scenarios, a SHA-based multiplexed cartridge with a small, portable signal reader may be employed. Commercial products of ready-to-use kits or devices meeting such requirements have been actively developed. For instance, the aforementioned CARLA kit and a portable SHA-based, multiple genera-targeted device (under development) adapted from a biosensor system described in Bickman et al. [63], both of which use immunoassays for signal detection and/or quantitation.

For the third scenario, the array-formatted SHA integrated into an ESP has been deployed for near real-time, autonomous field monitoring of marine bacterioplankton, including marine cyanobacteria [14,29,30,31,41,42,54]. An ESP is an electromechanical/fluidic system that collects discrete field water samples and detects target rRNAs present in the crude homogenate of sampled organisms [29,30,41]. The SHA process in the analytical module of the ESP can take real-time measurements of target bloom species, such as *Pseudo-nitzschia australis*, *Heterosigma akashiwo*, *Alexandrium catenella*, and *Synechococcus* spp. [29,30,41].

### 3.2. Improving SHA for More Convenient and Broader Applications for In-Situ cHAB Monitoring

In the foreseeable future, we anticipate much broader SHA applications for in situ cHAB detection and monitoring. Meanwhile, there exists substantial room for improvements in SHA’s throughput, sensitivity, specificity, portability, and species coverage. High throughputs may be achieved if a high-density microarray spotted with thousands of capture probes (replacing currently used 96-well plates or low-density arrays) is coupled with an automated imaging system for signal acquisition, processing, and analysis. An increase in throughput can also reduce costs in reagents, supplies, and labor. Sensitivity and specificity may be improved through a combination of increased in-field nucleic acid extraction efficiency, better design of capture and signal probes, more sensitive probe labelling, enhanced signal detection/imaging processing, and optimized hybridization conditions. For instance, a ten-fold increase in *Legionella* detection sensitivity was observed after moving a signal probe directly next to the capture probe [24]. Expansion of target cyanobacterial coverage is dependent on the availability of more cyanobacteria genomes and species or genus-specific gene sequences that have a high discriminatory power, which can be found in publicly accessible databases, such as CyanoBase (http://genome.microbedb.jp/cyanobase/) (accessed on 20 July 2022) and BioCyc (https://algae.biocyc.org/) (accessed on 20 July 2022).

### 3.3. Closing Remarks

As cHAB events increase in scale, severity, frequency, and duration around the world, rapid and accurate detection and monitoring tools have become critically essential for regulatory and management decision-making. SHA is one of the frequently employed molecular techniques for its significant advantages over other conventional tools, especially when applied to in situ, real-time field survey or monitoring. With more attention on technical development, improvement, and refinement, SHA-based technologies are believed to gain popularity in detecting and characterizing HAB-forming cyanobacteria in freshwater, brackish water, and marine water bodies.

## Figures and Tables

**Figure 1 biosensors-12-00640-f001:**
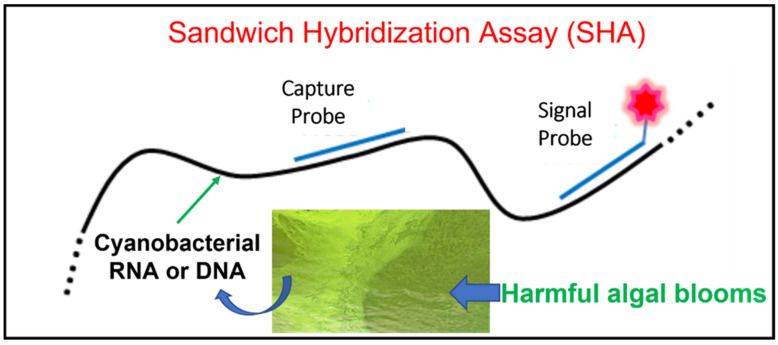
Schematic representation of sandwich hybridization assay (SHA) application to detection of harmful algal blooms-forming cyanobacteria. A target cyanobacterial nucleic acid molecule is “sandwiched” between a capture oligonucleotide probe immobilized on a solid support and a signal oligonucleotide probe labeled with a detection marker.

**Table 1 biosensors-12-00640-t001:** Representative applications and methodological variations of sandwich hybridization assays reported in the literature.

Reference	Solid Support	Capture Probe	Signal Probe	Detection Method	Washing Buffer	Hybridization Buffer	Incubation Time	Detection Limit	Detection Target
[12] Rautio et al. 2003	Streptavidin-coated magnetic beads	3′ Biotin labeled RNA	3′ DIG-labeled RNA	Plate fluorescence reader; Anti-DIG Fab fragment-AP conjugate + BBTP	50 mM Tris, 150 mM NaCl, 0.3% Tween 20	5 × SSC, 0.5% SDS, 0.02% Ficoll, 0.02% PVP, 0.02% BSA, 20% deionized formamide, 4% dextran sulfate	Denaturation: 65 °C 3–5 min	2 fmol (1.2 × 10^9^ molecules)	Yeast (*Saccharomyces cerevisiae*) SUC2 mRNA and 18S rRNA
							Hybridization: 50 °C 30 min		
							Bead immobilization: 37 °C 30 min		
							Conjugation: 20 °C 30 min		
							Substrate application: 37 °C 20 min		
[26] Feuillie et al. 2011	Streptavidin-coated magnetic beads	3′ Biotin labeled DNA	5′ Rhodamine 6G labeled DNA	Surface Enhanced Resonance Raman Scattering (SERRS)	0.25 × SSC, 0.5% Tween 20	4 × SSC, 0.05% Tween 20	Denaturation: 99 °C 10 min	1 fmol	Chamois (*Rupicapra rupicapra*) DNA
							Hybridization: 55 °C 3 h		
							Bead immobilization: 20 °C 30 min		
							Elution: 95 °C 20 min		
[15] Goldman et al. 2013	None (micelle drag tag-containg running buffer)	γ-carbon modified PNA (γPNA) amphiphile labelled with FITC	Cy5-labeled for DNA target or YOYO-1-stained duplex for RNA target	Capillary Electrophoresis equipped for laser-induced fluorescence detection	None	1 × TBE (Tris/Borate/EDTA)	Hybridization: 95 °C 5 min, cooled to room temperature in 1 h	Single base mismatch discrimina-tion (10 nm or 2 × 10^11^ molecules)	Short 22-nucleotide RNA or DNA
							YOYO-1 staining: 1 h at room temperature		
[17] Thieme et al. 2008	Streptavidin-coated magnetic beads	5′ Biotin labeled DNA + 2 unlabeled DNA helper probes	3′ DIG-labeled DNA	Microplate fluorescence reader; Anti-DIG Fab fragment-AP conjugate + BBTP	1 × SSC, 0.01% SDS in DEPC-treated water	5 × SSC, 20% formamide, 3% dextran sulfate, 0.2% Tween 20, 0.02% Ficoll 400, 0.02% PVP, 1% blocking reagent in 100 mM maleic acid with 150 mM NaCl, all mixed in DEPC-treated water	Plate incubation: 50 °C 5 min	<1 fmol of RNA per well	*Escherichia coli* mRNA
							Hybridization: 50 °C 30 min		
							Immobilization: 50 °C 30 min		
							Wash 1: 50 °C 2 min		
							Wash 2: 30 °C 2 min (twice)		
							Conjugation: 30 °C 30 min		
							Substrate application: 37 °C 20 min		
[25] Zhang and Appella 2007	DNA-BIND^®^ 96-well plate	PNAα covalently attached to plate	Biotin-labeled PNAβ	Plate reader; avidin-HRP + TMB	Wash 1 and 3: PBS; Wash 2: 0.1% SDS in 0.1 × SSC	Hybridization: 0.15 M NaCl Blocking buffer: 3% BSA and 25 mM lysine in 50 mM Na_2_HPO_4_/NaH_2_PO_4_, 1 mM EDTA	Attach PNAα to plate: 37 °C 1 h	10^−5^ fmol of DNA	*Bacillus anthracis* DNA (Anthrax)
							Wash 1: 33°C 1 min (3 times)		
							Blocking: 37°C 30 min		
							Hybridization: 45 °C 3 h		
							Wash 2: 33 °C 30 min (twice)		
							Blocking: 37°C 30 min		
							Conjugation: 37 °C 30 min		
							Wash 3: 37 °C 1 min (3 times)		
							Substrate application: 37 °C 20 min		
[14] Goffredi et al. 2006	Biotin-coated polystyrene prongs	5′-Biotinylated DNA, conjugated to streptavidin	Double DIG-labeled DNA at both 5′ and 3′ ends	Plate reader; anti-DIG-HRP + TMB-ELISA; robotic workstation	50 mM Tris, 150 mM NaCl, 0.05% Tween 20	2M GuSCN, 50 mM Tris, 10 mM EDTA, 0.5% Tween 20, pH 8.6	25 to 30 °C for all steps	5 larvae/mL of lysate	Barnacle 18S rRNA
							Attach capture probe to prong: 8 min		
							1st Hybridization (capture): 8 min		
							2nd Hybridization (signal): 8 min		
							1st Wash: 2 min		
							Antibody application: 5 min		
							2nd Wash: 2 min (twice)		
							Substrate application: 5 min		
[16] Clancy et al. 2017	Microarray glass slide	DNA covalently attached to the microarray glass slide	5′ Cy5-labeled DNA	Microarray scanner	Wash 1: 2 × SSC, 0.5% SDS;	Solution-phase pre-hybridization: 2 × SSC, 0.5% SDS, 1 µM reporter probe	Pre-hybridization: 30 °C 20 min	1 pM or 0.03 fmol of miRNA	Breast cancer related microRNA
							Slide preheating: 30 °C 10 min		
					Wash 2: 2 × SSC;		Hybridization: 30 °C 1 h		
					Wash 3: 0.2 × SSC		Wash 1: 30 °C 10 min		
							Wash 2&3: room temp 10 min		
[21] Zammatteo et al. 1997	Amine-grafted magnetic beads or polystyrene plate	DNA covalently attached to beads or plates	Biotinylated or radiolabeled DNA	Liquid scintillation counter for radio-labeled probe or plate reader (+ HRP-streptavidine + TMB) for biotinylated probe	Radiolabeled: 0.1 × SSC	4 × SSC, 10 × Denhart, 200 µg/mL DNA salmon sperm	Hybridization for both probe types: 60 °C for 2 h	0.03 fmol of HCMV DNA	Human cytomegalovirus (HCMV) DNA
					Biotinylated: buffer 1: 100mM maleic acid, pH 7.5, 150 mM NaCl, 0.3% Tween 20; blocking buffer: 100 mM maleic acid, pH 7.5, 150 mM NaCl, 0.1% Gloria milk powder; buffer 2: 100 mM maleic acid, pH 7.5, 150 mM NaCl		Biotinylated: streptavidin-peroxidase diluted in blocking buffer 23 °C for 45 min and TMB incubated 10 min in the dark		
[27] Scholin et al. 1996	Nylon beads	DNA conjugated to beads	5′ Biotin-labeled DNA	Visual inspection or photography	50 mM Tris HCI, 10 mM EDTA, 100 mM NaCl, 1% (*v*/*v*) SDS, 1% (*v*/*v*) N-Iauryl sarcosine, pH 8.0	Hybridization Buffer I: 100 mM Tris, 17 mM EDTA, 8.35% formamide, 5 M GuSCN, pH 7.5; Hybridization Buffer II: 100 mM Tris, 17 mM EDTA, 8.35% formamide, 3 M GuSCN, pH 7.5	Primary hybridization (target to bead): 23–25 °C for 30 min	Not reported	*Pseudo-nitzschia australis* large subunit (LSU) rRNA
							Secondary hybridization (signal probe to target): 23–25 °C 30 min		
							Washing: 23–25 °C 2 min (2X)		
							Conjugation: 23–25 °C 30 min		
							Substrate application: 23–25 °C 30 min		

**Table 2 biosensors-12-00640-t002:** Published studies of SHA application to cyanobacteria detection and quantification.

References	Solid Support	Detection Instrument & Method	Target Genus/Group	Target Gene	Capture Probe	Signal Probe
[43] Matsunaga et al. 2001	BMP (Bacterial Magnetic Particle)	Luminometer; Immunosorbent method: anti-DIG-AP used for signal detection after addition of the CDP-Star™ Substrate with Emerald-II™ Enhancer	*Anabaena*, *Microcystis*, *Nostoc*, *Oscillatoria*, *Synechococcus*	16S rRNA amplicon	Anabaena1 562–579 nt ABACWWACAATGCCACCT; Anabaena2 647–666 nt CCAGGAATTCCCTCTGCCC; Microcystis 585–604 nt TTAAGCAACCTGATTTGA; Nostoc1 569–587 nt ACAGCAGACTTACATTG; Nostoc2 628–636 nt ACGTACTCTAGCTATG; Oscillatoria 802–823 nt ACAGGCHACACCTAGTCTCCATC; Synechococcus 575–593 nt RGGCTTTGACARCAGACT	CYA-781R: 781–800 nt GACTACTGGGGTATCTAATCCCATT
[44] Matsunaga et al. 2001	BMP and MAG-microarray	Fluorescence stereomicroscope; Immunosorbent method: anti-DIG-AP used for signal detection after additon of the AttoPhos^®^ AP substrate	ditto	ditto	ditto (R = A or G; Y = C or T; W= A or T; K = G or T; M= A or C; S = G or C; H = A, C or T; V = A, C, or G; D = A, G or T; B = T or G; N = A, C, G or T)	ditto
[46] Castiglioni et al. 2004	Microarray spotted with universal “zip code” probes	ScanArray 4000 laser-scanning system (Cy3 with λ_ex_ = 543 nm;λ _em_ = 570 nm)	19 cyanobacterial groups	16S rRNA amplicons	Group-specific discriminating probes labeled with Cy3 dye at the 5’ end (see [46] for sequences)	Genus-specific common probes phosphorylated at the 5’ end and carrying a czip code at the 3’ end (see [46] for sequences)
[41] Preston et al. 2009	96-well plate or membrane array in Environmental sample processor (ESP)	Robotic processor/plate reader (A_450_) or CCD camera with digital image analysis system; colorimetric method: anti-DIG-HRP + HRP substrate	*Synechococcus* CCMP 1334	16S rRNA (in vitro transcripts or extracted RNA)	Picophyto496: 5’-Biotin-[C9 x 3]-GGCACGGAATTAGCCGWGGCTTA-3’	EUB338: 5’-DIG-[C9]-GCWGCCWCCCGTAGGWGT-[C9]-DIG-3’; Univ519ab: 5’-DIG-[C9]-TTACCGCGGCKGCTGGCAC-[C9]-DIG-3’
[47] Zhu et al. 2012	Magnetic beads modified with isothiocyanate groups	Cary 50 UV-Vis Spectrophotometer (A_405_)	*Microcystis*	*PC-IGS* amplicon	TF: 5’-GCAATAAGTTTCCTACGG-NH_2_	TR: 5’biotin-GGTATCTCCCAATAATCT-3’
[48] Zhu et al. 2012			*Microcystis*	*PC-IGS* amplicon	TF: 5’-GCAATAAGTTTCCTACGG-NH_2_	TR: 5’biotin-GGTATCTCCCAATAATCT-3’
	Immunosorbent method: Alkaline phosphatase-streptavidin + enzymatic substrate p-nitrophenyl phosphate sodium		*Microcystis*	*mcyJ* amplicon	TJF: 5’-CCAACCTTCCACCGGGCTGCA-NH_2_	TJR: 5’biotin-CGACCCACTCTAGGCAAACAATC-3’
[50] Dearth 2017; [51] Dearth et al. 2022	Streptavidin-coated 96-well plate	BioTek Synergy HT Plate reader (A_450_) & Affirm robotic processor; colorimetric method	*Microcystis*	16S rRNA (extracted RNA)	MIC593: 5’ biotin-[C9 x 3]-AACCTGATTTGACGGCAGACTTGGCTGA-3’	EUB338: 5’-DIG-[C9]-GCWGCCWCCCGTAGGWGT-[C9]-DIG-3’

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
