# Peer review of "Sandwich Hybridization Assay for In Situ Real-Time Cyanobacterial Detection and Monitoring: A Review"

_biosensors, 2022, doi:10.3390/bios12080640_

Round 1

Reviewer 1 Report

Comments and Suggestions for Authors

Manuscript review report "Sandwich Hybridization Assay for In Situ Real-Time Cyano- 2

bacterial Detection and Monitoring: A Minireview" by   Ping Gong, Anna K. Antrim, Sarah R. Bickman, Emily G. Cooley and Seung Ho Chung.

This paper is devoted to describing the issue of new sandwich hybridization test for real-time detection of cyanobacteria-2. This is of great interest from the point of view of environmental protection and human health protection. As is known, some species of cyanobacteria produce many poisonous substances The authors have presented sandwich hybridization for a new detector for in situ detection and monitoring of cyano-2 bacteria in a very interesting and clear way The tables are very clear despite the abundant content.   Thus, the work is very promising and can be recommended for publication in present form.

·       I would suggest swapping the reference designation - from " 1 " to [1].

·       I would also suggest adding „doi” numbers in the references.

Good Luck ?

Author Response

Response to Reviewer 1 Comments

Point 1: I would suggest swapping the reference designation - from " 1 " to [1].

Response 1: We have made changes throughout the manuscript as suggested by this reviewer. Very much appreciate the comment!

Point 2:   I would also suggest adding „doi” numbers in the references

Response 2: We checked the journal's Instruction for Authors and the latest papers published in Biosensors. But we did not find any requirements for adding doi info for cited references. The journal or publisher may have an automatic system to add [Google Scholar] [CrossRef] [PubMed] links for each reference. So, we choose to keep it as it is and leave it to the publisher or journal editor to make the final call.

Reviewer 2 Report

The authors reviewed sandwich hybridization assay (SHA) as an accurate, real-time and field-portable device for the detection of cyanobacterial harmful algal blooms (cHABs) in environmental systems. The authors have thoroughly detailed the development history of SHA, summarized various types of SHA in the literatures, and discussed the advantages and disadvantages of using SHA for monitoring cHABs. Generally, this minireview is well-organized and presented in a very logical manor. I would recommend it for publication in Biosensors after addressing the following concerns:

1). In at least one case (Page12, Line 286), the weblink in the manuscript does not direct the readers to the correct page. The authors need to double check and provide the corresponding link addresses.

2). In the References section, reference 3 in particular, it shows that the page could not be found. The authors need to update the address, and include the date when it was accessed for clarity.

Author Response

Response to Reviewer 2 Comments

Point 1. In at least one case (Page12, Line 286), the weblink in the manuscript does not direct the readers to the correct page. The authors need to double check and provide the corresponding link addresses. 

Response 1. We have double checked all the weblinks (including https://www.microbia-environnement.com/en/technology/). They all worked. The Microbia Environnement website may not be accessible if one's search engine has set up certain firewalls.

 Point 2. In the References section, reference 3 in particular, it shows that the page could not be found. The authors need to update the address, and include the date when it was accessed for clarity.

Response 2. Again, we were able to successfully retrieve the documents cited in both Refs. 3 and 50. We added retrieving and access date as suggested by the reviewer.

Reviewer 3 Report

As cyanobacterial harmful algal bloom events increase in scale, severity, frequency and duration around the world, rapid and accurate detection and monitoring tools have become critically essential for regulatory and management decision-making. The manuscript is well organized and interesting. Minor revision is suggested and the comments are listed as below.

1.     There are too many keywords. 6 keywords would be enough.

2.     A wide variety of approaches and technologies have been developed for the detection and monitoring of various bacteria. Some closely related references are suggested to be cited for broad readers, for example A review on conversion of crayfish-shell derivatives to functional materials and their environmental applications; Preparation and properties of cellulose nanocomposite fabrics with in situ generated silver nanoparticles by bioreduction method.

3.     There should be a space between the number and the unit, for example “69°C and 74°C” in line 97 and other places.

4.     “linear range” is an important parameter for biosensors which is suggested to be added in table 1 for comparison.

5.     The references are too old. More references published recently are suggested to be cited.

Author Response

Response to Reviewer 3 Comments

Point 1: There are too many keywords. 6 keywords would be enough

Response 1: According to Instructions for Authors, three to ten pertinent keywords need to be added after the abstract. We listed 8 relevant keywords (some more specific like "sandwich hybridization assay" while others more general like "water quality"), which we think meet the journal's requirements in terms of keyword number and relevance.

Point 2: A wide variety of approaches and technologies have been developed for the detection and monitoring of various bacteria. Some closely related references are suggested to be cited for broad readers, for example A review on conversion of crayfish-shell derivatives to functional materials and their environmental applications; Preparation and properties of cellulose nanocomposite fabrics with in situ generated silver nanoparticles by bioreduction method.

Response 2: One of the main objectives of this review article is to provide an in-depth coverage of SHA-based techniques for environmental microbial sample detection. To stay focused on this topic, we did not attempt to discuss other techniques (e.g., qPCR, ELISA) in much detail. Instead, we briefly mentioned this (see the second paragraph in page 4 and below) and provided a source of review article (reference #10) for such info.

A wide variety of approaches and technologies have been developed for the detection and monitoring of HABs-forming cyanobacteria, ranging from microscopic enumeration, analysis of Chlorophyll a, ATP and phycocyanin, quantitative polymerase chain reaction (qPCR), next-generation sequencing (NGS), enzyme-linked immunosorbent assays (ELISA), and high-pressure liquid chromatography (HPLC), to hyperspectral imaging, remote sensing, automated cell imaging systems, and machine learning [10].   

As for the two references "A review on conversion of crayfish-shell derivatives to functional materials and their environmental applications" and "Preparation and properties of cellulose nanocomposite fabrics with in situ generated silver nanoparticles by bioreduction method" mentioned by this reviewer, we downloaded and read them. Unfortunately, we found both articles are not directly related to the topic of our manuscript.

Point 3: There should be a space between the number and the unit, for example “69°C and 74°C” in line 97 and other places.

Response 3: We thank the reviewer for this comment and have corrected all such instances throughout our manuscript.

Point 4:  “linear range” is an important parameter for biosensors which is suggested to be added in table 1 for comparison.

Response 4: We agree with the reviewer on the importance of this parameter that indicates the robust range of quantitative target detection. Unfortunately, most of the studies we came across did not report the linear range of detection. Most of them only report detection limits. So, we had a column in Table 1 to show and compare the lowest detection limit (sensitivity). 

Point 5:  The references are too old. More references published recently are suggested to be cited.

Response 5: We believe that we have surveyed the majority of relevant literature that spans a time frame of >3 decades (1990 to present). Statistics shows that we cited 11 references prior to the year 2000, 28 between 2000 and 2010, and 24 from 2011 to 2022, which indicates a good balance of coverage. Especially, our review covered the most recent developments in the topic area. We would be very much appreciative if the reviewer could provide us any additional latest references that we may have missed. We would be more than happy to include them in our manuscript.